# Reverse-engineered models reveal differential membrane properties of autonomic and cutaneous unmyelinated fibers

**Brandon J. Thio**[1], **Nathan D. Titus**[1], **Nicole A. Pelot**[1], **Warren M. Grill**[1,2,3,4]*

**1** Department of Biomedical Engineering Duke University Durham, North Carolina, United States of America, **2** Duke University, Department of Electrical and Computer Engineering, Durham, North Carolina, United States of America, **3** Duke University School of Medicine, Department of Neurobiology, Durham, North Carolina, United States of America, **4** Duke University School of Medicine, Department of Neurosurgery, Durham, North Carolina, United States of America

* warren.grill@duke.edu

**Data Availability Statement:** All code and data for the C-fiber models, for the PSO framework, and to

## Abstract

Unmyelinated C-fibers constitute the vast majority of axons in peripheral nerves and play key roles in homeostasis and signaling pain. However, little is known about their ion channel expression, which controls their firing properties. Also, because of their small diameters (~ 1 μm), it has not been possible to characterize their membrane properties using voltage clamp. We developed a novel library of isoform-specific ion channel models to serve as the basis functions of our C-fiber models. We then developed a particle swarm optimization (PSO) framework that used the isoform-specific ion channel models to reverse engineer C-fiber membrane properties from measured autonomic and cutaneous C-fiber conduction responses. Our C-fiber models reproduced experimental conduction velocity, chronaxie, action potential duration, intracellular threshold, and paired pulse recovery cycle. The models also matched experimental activity-dependent slowing, a property not included in model optimization. We found that simple conduction responses, characterizing the action potential, were controlled by similar membrane properties in both the autonomic and cutaneous C-fiber models, but complicated conduction response, characterizing the afterpotentials, were controlled by differential membrane properties. The unmyelinated C-fiber models constitute important tools to study autonomic signaling, assess the mechanisms of pain, and design bioelectronic devices. Additionally, the novel reverse engineering approach can be applied to generate models of other neurons where voltage clamp data are not available.

## Author summary

Computational models of axons play an important role in studying neural signaling and developing therapeutic electrical stimulation devices. While robust models of large myelinated axons exist, models of unmyelinated C-fibers do not adequately reproduce experimental conduction responses. C-fibers constitute the vast majority of axons in peripheral nerves and play key roles in homeostasis and signaling pain, but because of their small

reproduce the Figs in this paper are available at
https://gitlab.oit.duke.edu/bjt20/thio_cfiber.

**Funding:** This work was supported by NIH SPARC
OT2 OD025340 to WMG and NIH R01 NS126376
to WMG. The funders had no role in study design,
data collection and analysis, decision to publish, or
preparation of the manuscript.

**Competing interests:** The authors have declared
that no competing interests exist.

diameters ($\sim 1$ μm), it has not been possible to characterize their membrane properties using voltage clamp. We used particle swarm optimization and a novel library of isoform-specific ion channel models to reverse engineer C-fiber membrane properties from measured autonomic and cutaneous C-fiber conduction responses. Our C-fiber models reproduced experimental conduction velocity, chronaxie, action potential duration, intracellular threshold, paired pulse recovery cycle, and activity dependent slowing. The models constitute important tools to study autonomic signaling, assess the mechanisms of pain, and design bioelectronic devices. Additionally, the novel reverse engineering approach can be applied to generate models of other neurons where voltage clamp data are not available.

## 1 Introduction

Unmyelinated C-fibers constitute the vast majority of axons in the peripheral nervous system [1–3], yet little is known about the expression of ion channels that determine their firing properties. Subpopulations of C-fibers show substantial differences in electrophysiological properties [4], and patch clamp studies of neuronal somata confirm differences in expression of ion channel isoforms between C-type neurons [5]. However, the exceedingly small size of unmyelinated axons (0.5–1.5 μm) has precluded comparable axonal voltage clamp studies. Axonal physiology differs substantially from somatic physiology [6,7], and differential expression of ion channel isoforms may account for differences between somatic and axonal physiology [8]. Therefore, we used computational optimization to reverse engineer the expression of ion channel isoforms necessary to generate the emergent electrophysiological properties of autonomic and cutaneous C-fibers.

Cable models of axons use differential equations to represent nonlinear ion channels with properties obtained from voltage clamp experiments [9]. This approach led to a robust model of myelinated axons that reproduced a broad range of experimental characteristics [10]. However, comparable voltage clamp data are not available for C-fibers, and prior C-fiber models were parameterized to match specific experimental characteristics [11–14]. We developed new C-fiber models that reproduce a broad range of experimental responses (Table 1), and these models are important new tools for studying autonomic signaling, pain signaling, and responses to therapeutic electrical stimulation.

Voltage-gated ion channels are represented mathematically in disparate ways across different studies [3]. Therefore, we developed a library of isoform-specific ion channel models that formed the basis functions for our optimization. Subsequently, we developed a multi-objective particle swarm optimization (PSO) framework [15] to identify key membrane parameters in autonomic and cutaneous C-fiber models that replicate experimental conduction responses. Finally, we validated our models using activity-dependent slowing (ADS) data that were not used during the optimization. Together our results provide validated models that can be used to interrogate C-fiber responses to stimuli and a new framework to create neuron models when patch clamp measurements are unavailable.

## 2 Results

### Library of Isoform-Specific Ion Channel Models

Many existing neuron models include disparate representations of the same ion channel isoform, use one ion channel model to represent multiple isoforms, or include ion channel

**Table 1. Conduction responses of autonomic [13,14] and cutaneous [11,12] C-fiber models and experiments.** For the four published models, the values in each cell are the responses for 1 μm fibers. For the "Target" columns, the values are the lower and upper bounds for that target response used in the optimization, as measured in the experiments of the "References" columns ±10% (autonomic) and ±0% (cutaneous). Each simulation was conducted at the indicated temperature. The bolded values represent model responses that are not within the experimental range. NA: not applicable.

| Conduction Responses | Autonomic C-fiber | | | | Cutaneous C-fiber | | | |
| --- | --- | --- | --- | --- | --- | --- | --- | --- |
| | Schild 1994 | Schild 1997 | Target (Temp.) | References | Sundt 2015 | Tigerholm 2014 | Target (Temp.) | References |
| Conduction Velocity (m/s) | 0.512 | **0.294** | 0.45 1.76 (37˚C) | [23] [24] | **0.488** | 0.776 | 0.5 1.5 (37˚C) | [25] |
| Chronaxie of Strength-Duration Curve (ms) | 1.069 | 1.310 | 0.75 1.5 (37˚C) | [26] [27] | **1.619** | 0.813 | 0.5 1.5 (33˚C) | [28, 29] |
| Action Potential Duration (ms) | **7.69** | **5.23** | 1.35 3 (37˚C) | [30] [31] | 1.62 | **6.65** | 1.5 2.6 (24˚C) | [32] |
| Duration of Refractory Period (ms) | >**47** | >**250** | >13 (37˚C) | [33] | ---- | ---- | ---- | |
| Magnitude of Supernormal Epoch (Multiple of Threshold) | **0.99** | **1.03** | 0.7 0.9 (37˚C) | [33] | 1.085 | **1.152** | 0.9 1.1 (33˚C) | [34] |
| Magnitude of Subnormal Epoch (Multiple of Threshold) | **0.94** | **1.01** | 1.02 1.15 (37˚C) | [33] | **0.99** | **0.99** | 1.1 1.3 (33˚C) | [34] |
| Timing of Crossover from Supernormal to Subnormal Epoch (ms) | **NA** | **NA** | 50 100 (37˚C) | [33] | ---- | ---- | ---- | |
| Intracellular Current Threshold, 10 ms Pulse (nA) | ---- | ---- | ---- | | 0.02 | 0.08 | 0.015 0.15 (24˚C) | [32] |

models that do not adequately match data from patch clamp experiments (S1–S11 Figs). Therefore, we developed a library of 12 isoform-specific voltage-sensitive ion channel models by fitting experimental data from multiple sources to Hodgkin-Huxley-style equations [16] (Figs 1 and S1–S11 and S1 Table). We chose to only include voltage gated ion channel isoforms that were present in previous C-fiber models or were shown, in experimental characterization, to be present in peripheral C-fibers [3]. These validated ion channel models served as the basis functions of our C-fiber models.

## Particle Swarm Optimization Identified Model Parameters That Reproduce Experimental C-fiber Responses

We developed a PSO algorithm (Fig 2) to identify model parameters—maximum ion channel conductances, maximum pump currents, intracellular resistivity, and resting transmembrane potential—that reproduced experimental C-fiber conduction responses (Table 1). Each complete set of model parameters constituted a "particle". The PSO initialized a population of particles with random values, and the fitness of each particle was evaluated and scored according to the target performance criteria, including the conduction velocity, strength-duration properties, action potential duration, refractory period, excitability recovery period, and intracellular threshold (Table 1). The PSO then adjusted the velocity of each particle to move through the bounded multi-dimensional parameter space. Updates to the velocities were driven by the parameter values of particles that best matched each performance criterion. The PSO iteratively evaluated and updated the population of particles until a particle that met all performance criteria was found or a maximum number of iterations elapsed (S12 Fig).

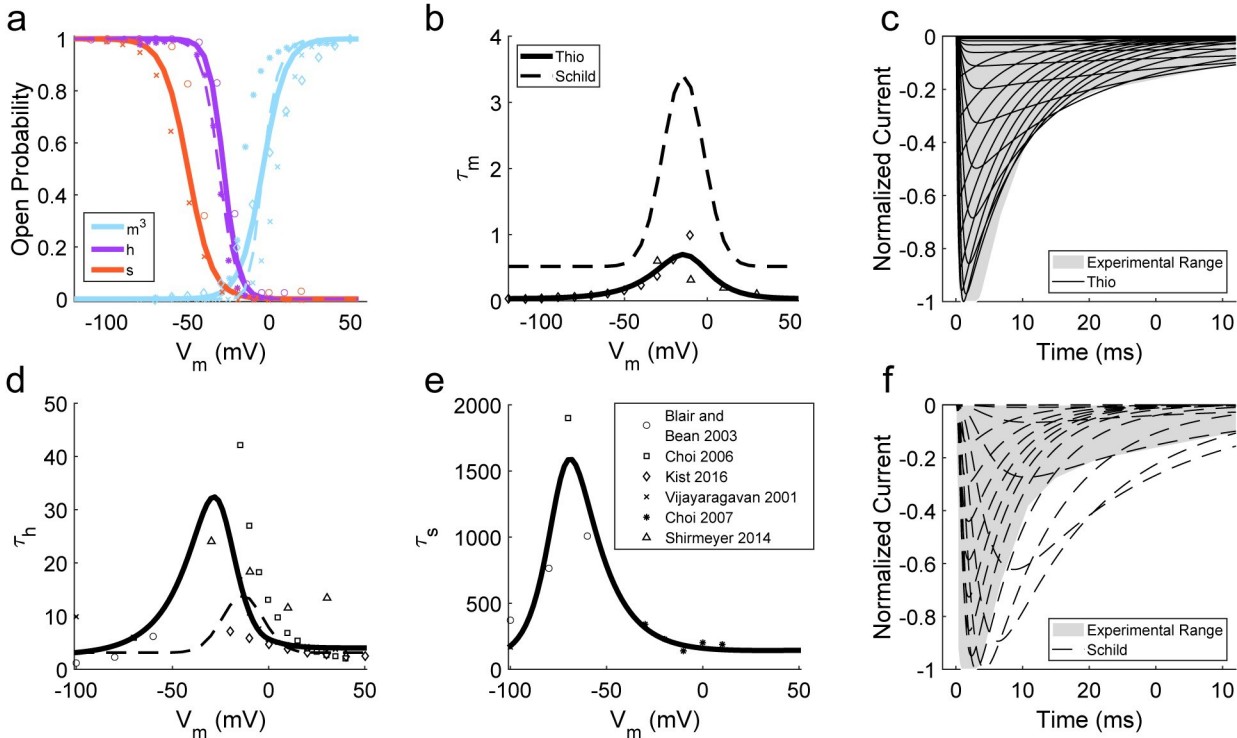

**Fig 1.** Comparison of our isoform-specific model of Na$_V$1.8 (solid line) and the Schild and Kunze Na$_S$ model (dashed line) [14] to experimental data (markers and grey shading) [17–22]: a) steady-state gating parameter values, b) activation time constant, d) fast-inactivation time constant, e) slow-inactivation time constant, c) novel Na$_V$1.8 model current responses to voltage clamps, and f) Schild Na$_S$ model current responses to voltage clamps.

We simulated 50 PSOs using autonomic C-fiber performance criteria and 20 PSOs using cutaneous C-fiber performance criteria (Table 1). Thirteen of 50 autonomic and 14 of 20 cutaneous PSOs identified a set of model parameters that reproduced all the target experimental responses. To evaluate further the 13 autonomic and 14 cutaneous candidate fiber models, we simulated their activity-dependent slowing responses (ADS); one autonomic and one cutaneous fiber model reproduced experimental ADS (Table 2).

## Optimized models reproduce experimental c-fiber responses better than other models

We simulated the responses of three diameters for each C-fiber model (0.5, 1, and 1.5 μm), and the optimized autonomic and cutaneous C-fiber models better replicated experimental responses than other published C-fiber models [11–14] (Fig 3). The optimized models reproduced all experimental response targets, as defined for the PSO. However, both Schild autonomic models only reproduced the experimental strength-duration characteristics, and there were pronounced differences compared to the experimental recovery cycle dynamics. For cutaneous fibers, the Sundt model did not consistently reproduce the experimental conduction speed or recovery cycle dynamics, and the Tigerholm model did not reproduce the experimental action potential duration or recovery cycle dynamics.

## Models reproduce experimental data not used in optimization

We evaluated the predictive power of our C-fiber models to reproduce experimental data that were not used in the optimization. We simulated ADS in the 13 autonomic and 14 cutaneous

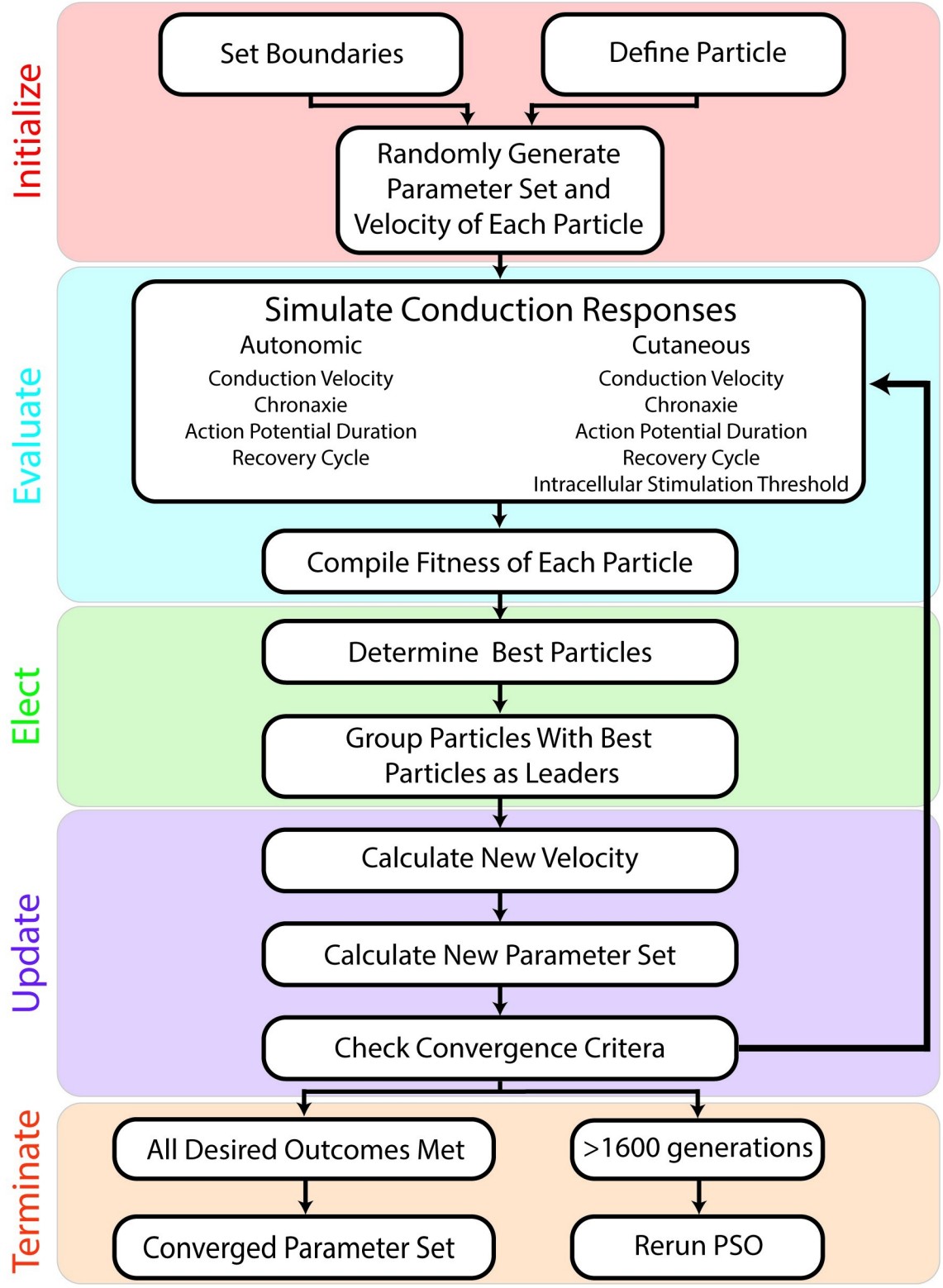

**Fig 2. Particle swarm optimization (PSO) algorithm to identify parameter values of model C-fiber axons and meet performance criteria.** The evaluation, election, and update steps are repeated until termination criteria are met.

**Table 2. Parameter values for optimized autonomic and cutaneous C-fiber models that also reproduced experimental activity-dependent slowing response.**

| Model Parameters | Autonomic PSO Fiber | Cutaneous PSO Fiber |
|---|---|---|
| $\bar{g}$, $Na_V1.7$ ($mS/cm^2$) | 36.81 | 35.66 |
| $\bar{g}$, $Na_V1.8$ ($mS/cm^2$) | 75.75 | 115.6 |
| $\bar{g}$, $Na_V1.9$ ($mS/cm^2$) | 0.376 | 0.504 |
| $\bar{g}$, $K_V1.4$ ($mS/cm^2$) | 0.024 | 0.044 |
| $\bar{g}$, $K_V2.1$ ($mS/cm^2$) | 5.337 | 327.2 |
| $\bar{g}$, $K_V3.4$ ($mS/cm^2$) | 0.289 | 1.786 |
| $\bar{g}$, $K_V7$ ($mS/cm^2$) | 2.864 | 0.003 |
| $\bar{g}$, $KCa_{BK}$ ($mS/cm^2$) | 0.156 | 2.016 |
| $\bar{g}$, $KCa_{SK}$ ($mS/cm^2$) | 0.006 | 0.755 |
| $\bar{g}$, $Ca_V1.2$ ($mS/cm^2$) | 0.004 | 0.188 |
| $\bar{g}$, $Ca_V2.2$ ($mS/cm^2$) | 9.546 | 0.361 |
| $\bar{g}$, HCN ($mS/cm^2$) | 2.789 | 0.106 |
| $\bar{g}$, $Na_{Leak}$ ($mS/cm^2$) | 0.110 | 0 |
| $\bar{g}$, $K_{Leak}$ ($mS/cm^2$) | 0.529 | 0 |
| $\bar{g}$, $Ca_{Leak}$ ($mS/cm^2$) | 0 | 0 |
| Max Current $Na_{pump}$ ($\mu A/cm^2$) | 0 | 5.131 |
| Max Current $K_{pump}$ ($\mu A/cm^2$) | 0 | 10.080 |
| Max Current $Ca_{pump}$ ($\mu A/cm^2$) | 0.069 | 0.049 |
| Max Current $NaCa_{exchanger}$ ($\mu A/cm^2$) | 0.210 | 9.242 |
| Max Current $NaK_{pump}$ ($\mu A/cm^2$) | 56.32 | 0.456 |
| Intracellular Resistivity (Ra) ($\Omega$-cm) | 23.11 | 27.51 |
| Rest Potential (mV) | -58.44 | -58.49 |

C-fiber models that met all performance criteria. The conduction velocity of an action potential is dependent on the activation history of the fiber, and continuous low frequency firing leads to slowed conduction in both autonomic and cutaneous C-fibers [4,35]. Therefore, we stimulated the model C-fibers at 0.5, 1, 2, or 4 Hz for 180 s and calculated the change in conduction velocity of each action potential relative to the conduction speed of the first spike. Only 1/13 autonomic and 1/14 cutaneous fibers reproduced experimental ADS (S13 Fig).

The autonomic C-fiber model matched the magnitude and time course of ADS in rat vagal C-fibers (Fig 4A; 1 μm autonomic C-fiber: $ADS_{2Hz}$ = 1.6% model vs. 1.9% experiment at t = 180 s) [35]. The cutaneous C-fiber model exhibited ADS comparable to low threshold mechano-responsive cutaneous C-fibers (Fig 4B; 1 μm cutaneous C-fiber $ADS_{2Hz}$ = 14% model vs. 13.8% experiment at t = 180 s) [4]. As well, the autonomic C-fiber models exhibited considerably less ADS compared to the cutaneous C-fiber models for all stimulation frequencies, consistent with experimental measurements [4,35].

A potential mechanism of ADS is intracellular accumulation of sodium leading to a reduction in the $Na^+$ current and subsequent decrease in conduction velocity [11]. The time course of ADS for both C-fiber models was coincident with changes in the intracellular $Na^+$ concentration consistent with the proposed mechanism (S14 Fig). Additionally, there was a larger increase in intracellular $Na^+$ concentration in the cutaneous C-fiber model than in the autonomic C-fiber model (22.7 mM vs. 1.3 mM maximum change), consistent with the larger ADS in the cutaneous model. While there were negligible changes in the steady state periaxonal $K^+$ concentrations compared to baseline, consistent with previous simulation studies [11], our cutaneous C-fiber model exhibited a transient decrease in ADS magnitude at ~5 s during 4 Hz

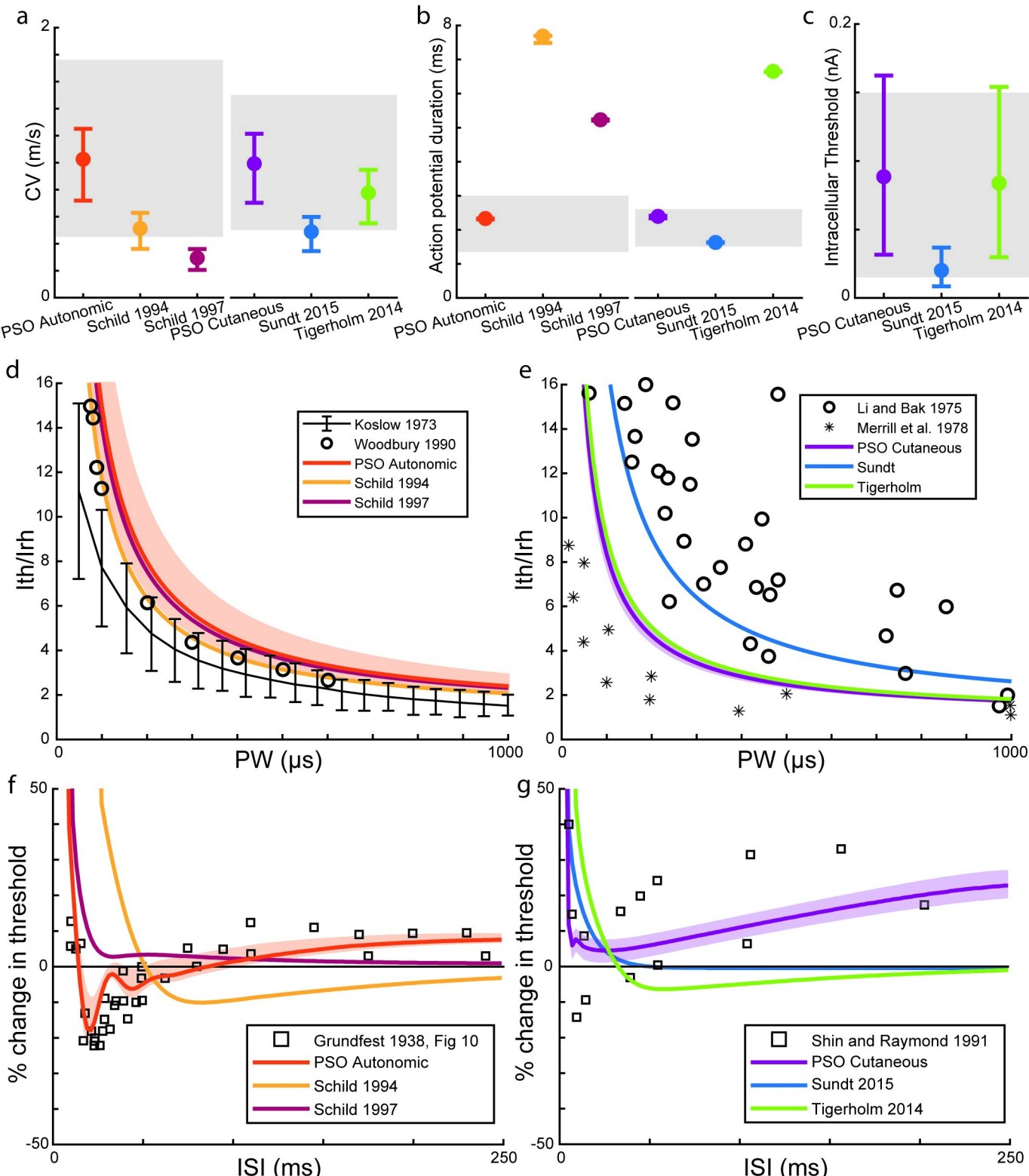

**Fig 3. Conduction responses in optimized C-fiber models (PSO Autonomic and PSO Cutaneous) and published C-fiber models compared to experimental recordings (Table 1).** a) Conduction velocity (left: autonomic, right: cutaneous), b) action potential duration (left: autonomic, right: cutaneous), c) intracellular threshold (cutaneous), d) strength-duration (autonomic), e) strength-duration (cutaneous), f) threshold recovery cycle (autonomic), and g) threshold recovery cycle (cutaneous). In panels a to c, experimental ranges are shown as grey shaded regions and the error bars and data points represent the conduction responses

for three simulated fiber diameters (center = 1 μm, range = [0.5 μm, 1.5 μm]). In panels d to g, the data points denote experimental data, and the shaded areas correspond to the range of conduction responses from the three simulated diameters; the lines for the previously published models correspond to the conduction responses for 1 μm fibers.

stimulation (Fig 4B). This decreases in ADS magnitude directly followed a transient increase in periaxonal $K^+$ concentration (S14 Fig). Therefore, while intracellular $Na^+$ concentration appears to be the major driver of ADS, periaxonal $K^+$ concentration may influence short-term (<10 s) ADS dynamics.

## Ionic currents vary greatly between autonomic and cutaneous C-fibers

To characterize further the mechanisms underlying the differences in excitability between autonomic and cutaneous C-fibers, we recorded the ionic currents contributing to the action potential and afterpotentials for both C-fiber models (Fig 5). $Na_V1.7$ and $Na_V1.8$ were the dominant $Na^+$ channels during the rising phase of the action potential, and $K_V2.1$ and $Na_V1.8$ were the dominant currents for the remaining duration of the action potential following the peak. While $K_V2.1$ and $Na_V1.8$ contributed to 75% of the total current at the peak of the autonomic C-fiber model's action potential, many other channels were also active ($K_V3.4$, $K_V7$, $Ca_V1.2$, and NaK Pump). In contrast, $Kv2.1$ and $Na_V1.8$ contributed to 98% of the total current at the peak of the cutaneous C-fiber model's action potential.

The ionic currents during the afterpotential differed greatly between the autonomic and cutaneous C-fiber models. The NaK pump dominated ionic currents in the autonomic fiber, while the NaK pump contributed minimally to the afterpotential in the cutaneous C-fiber. $K_V7$ was the dominant $K^+$ channel active during the afterpotential in the autonomic C-fiber model while $K_V2.1$ and $KCa_{SK}$ were the dominant $K^+$ channels in the cutaneous C-fiber model. $Na_V1.7$ and $Na_V1.8$ were active during the afterpotential for both the autonomic and cutaneous C-fiber models.

Despite using the same constituent ion channels, the ionic currents in the autonomic and cutaneous C-fiber models differed greatly during the action potential and afterpotential. This

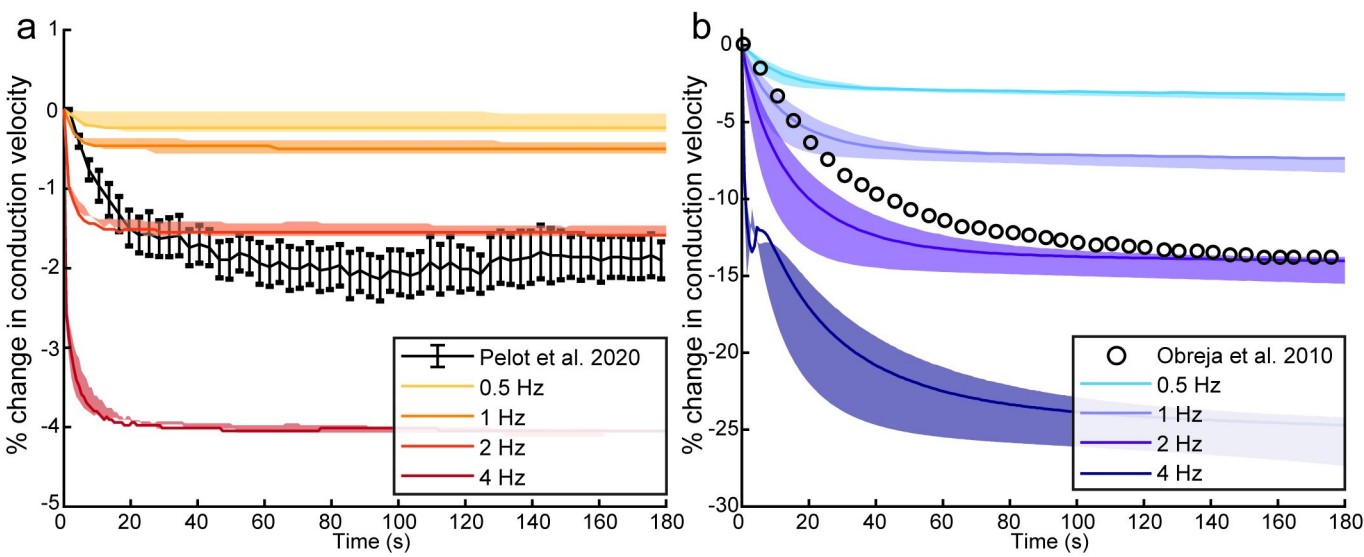

**Fig 4.** Activity-dependent slowing of action potential conduction in optimized C-fiber models with 0.5, 1, 2, and 4 Hz stimulation compared to experimental data with 2 Hz stimulation for a) autonomic [35] and b) cutaneous [4] C-fibers. Lines show data for 1 μm C-fiber models; shaded regions show range for 0.5 to 1.5 μm C-fiber models. Note: the % change in conduction velocity for a) autonomic and b) cutaneous C-fiber models differ in scale.

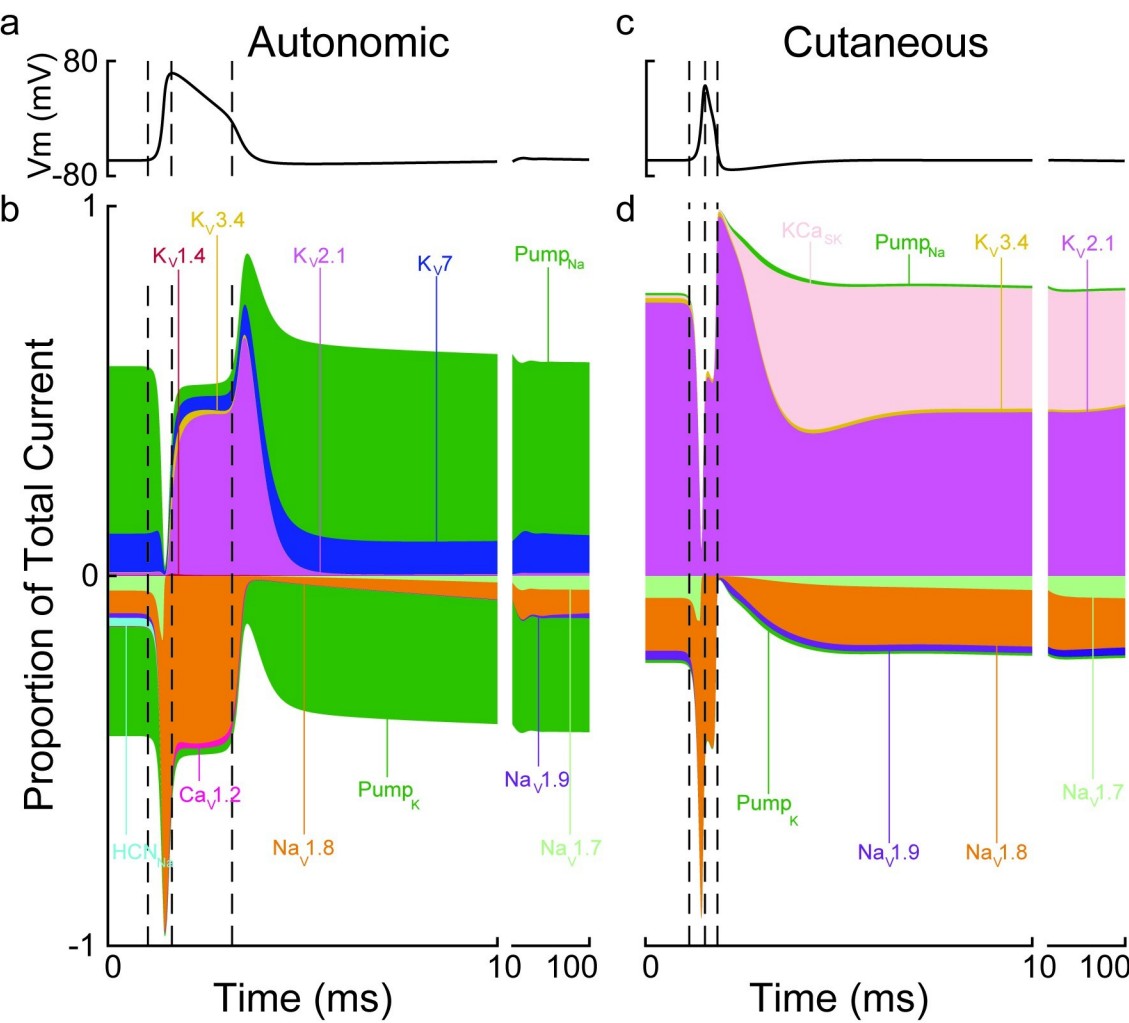

**Fig 5.** Transmembrane potential (a and c) and proportions of total ionic current (b and d) during the action potential and afterpotentials for the autonomic (left) and cutaneous (right) C-fiber models. In b and d, inward currents are shown as a negative proportion of the total current and outward currents are shown as a positive proportion of total current. The three vertical dashed lines correspond to the action potential initiation, action potential peak, and action potential shoulder. Only the dominant ion currents are visible, but all ion channels were included in the simulated models.

emphasizes the importance of understanding the expression of ion channels and shows the power of our PSO method for reverse engineering membrane properties using available electrophysiology data.

### Specific ion channels are necessary to model C-fiber conduction responses

To understand the influence of the membrane parameters on C-fiber excitability, we conducted sensitivity analyses by increasing or decreasing each parameter by 50% (Figs 6 and 7 and S2 and S3 Tables). The responses of both autonomic and cutaneous C-fiber models were most sensitive to changes in a subset of ion channels. Both models exhibited a >10% change in conduction velocity when we changed $Na_V1.8$ and Ra, in chronaxie (minimum pulse width that can evoke an action potential with twice the minimum activation current) when we changed $Na_V1.7$ and $Na_V1.8$, and in action potential duration when we changed $Na_V1.8$ and

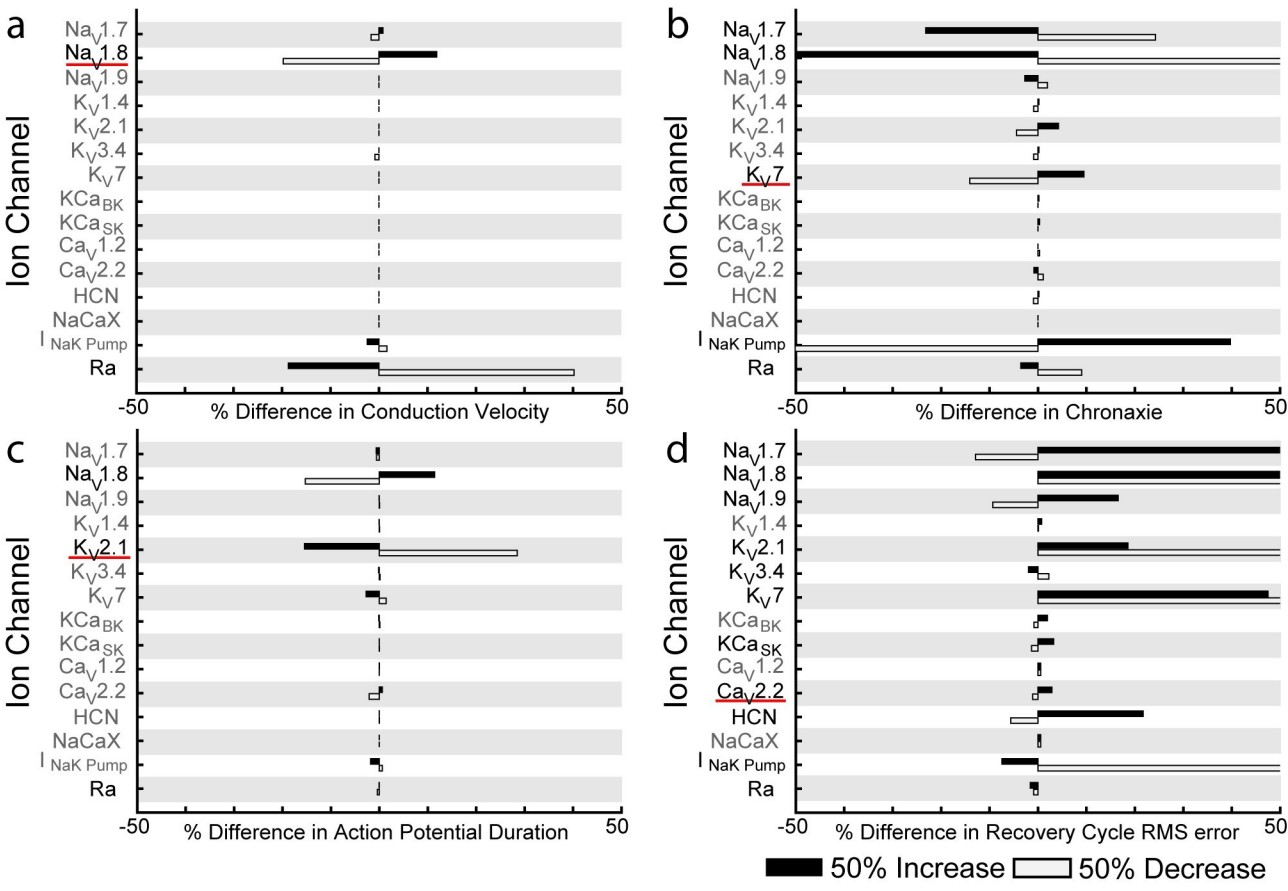

**Fig 6. Sensitivity of autonomic C-fiber conduction responses to model parameters.** Sensitivity of a) conduction velocity, b) chronaxie of the strength-duration curve, c) action potential duration, and d) recovery cycle root mean square (RMS) error compared to experimental data (see Methods) are shown for 50% changes in parameter values (max conductance, max pump current, or Ra) for 1 μm autonomic C-fiber models. Bolded ion channels and pumps are the minimum set of ion channels and pumps needed to reproduce the experimental conduction response. When the ion channels underlined in red were removed, the model no longer matched the experimental data.

$K_V2.1$ (Figs 6A–6C and 7A–7C). However, we observed differences in chronaxie sensitivity to changes in $K^+$ channels and the NaK pump: chronaxie changed by >10% by changing $K_V7$ and NaK pump of the autonomic model and by changing $K_V2.1$ for the cutaneous model (Figs 6B and 7B). Intracellular activation threshold of the cutaneous C-fiber model changed by >10% in response to changes in $Na_V1.7$, $Na_V1.8$, $K_V2.1$, and Ra (Fig 7E).

While only a few ion key ion channels controlled the conduction velocity, chronaxie, action potential duration, and threshold of the C-fiber models, many ion channels contributed to the recovery cycle dynamics. The recovery cycle error changed by >10% in response to changing $Na_V1.7$, $Na_V1.8$, $Na_V1.9$, $K_V2.1$, $K_V7$, HCN, and the NaK Pump for the autonomic C-fiber model, and in response to changing $Na_V1.8$, $K_V2.1$, $KCa_{SK}$, and $Ca_V1.2$ for the cutaneous C-fiber model (Figs 6D and 7D).

To reduce the complexity of the C-fiber models, we determined the minimum necessary set of ion channels to reproduce the target responses. We iteratively removed ion channels and pumps, from lowest to highest sensitivity, until each conduction response was no longer within experimental bounds. Conduction velocity for both autonomic and cutaneous C-fiber models only required the $Na_V1.8$ channel (Figs 6A and 7A). Action potential duration for both models, cutaneous threshold, and cutaneous chronaxie only required the $Na_V1.8$ and $K_V2.1$

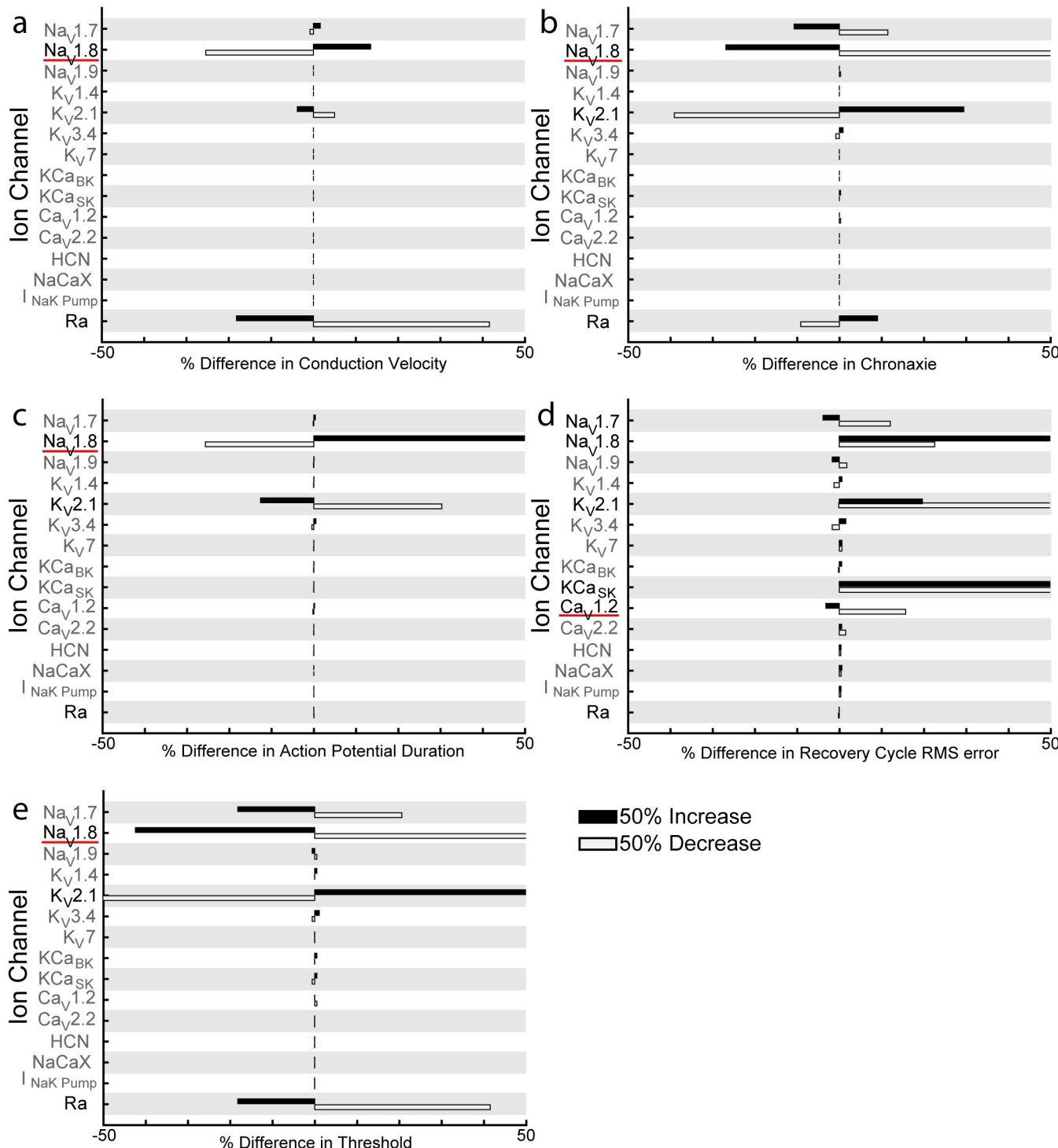

**Fig 7. Sensitivity of cutaneous C-fiber conduction responses to model parameters.** Sensitivity of a) conduction velocity, b) chronaxie of the strength-duration curve, c) action potential duration, d) recovery cycle root mean square (RMS) error compared to experimental data (see Methods), and e) intracellular activation threshold are shown for 50% changes in parameter values (max conductance, max pump current, or Ra) for 1 μm cutaneous C-fiber models. Bolded ion channels and pumps are the minimum set of ion channels and pumps needed to reproduce the experimental conduction response. When the underlined ion channels were removed, the model no longer matched the experimental data.

channels (Figs 6C, 7B,7C and 7E). Autonomic chronaxie required $Na_V1.7$, $Na_V1.8$, $K_V7$, and the NaK pump (Fig 6B). However, the recovery cycle dynamics for both fibers required many more ion channels and pumps (Figs 6D and 7D). These findings further demonstrate that generating models that reproduce action potential-based conduction responses is less challenging, but generating models that reproduce afterpotential-based conduction responses is much more so.

## 3 Discussion

C-fibers constitute the vast majority of peripheral afferents, but their small axon diameters (~1 μm) prevent voltage clamp to characterize their membrane properties. We developed isoform-specific ion channel models and a particle swarm optimization framework to reverse engineer models of autonomic and cutaneous C-fibers. Our optimized models reproduced a broad range of experimental conduction responses and exhibited ADS that matched experimental measurements, even though ADS was not part of the optimization. The resulting models constitute important tools to assess patterns of afferent activity under physiological and pathophysiological conditions as well as for analysis and design of bioelectronic devices. The novel reverse engineering approach and PSO framework can be applied to develop models of other neurons where electrophysiological characterization has been conducted, but where voltage clamp data are not available.

Ion channel isoforms are encoded by different genes and possess distinct electrophysiological properties [36]. However, many existing ion channel models combine characteristics from many isoforms into a single ion channel model [3]. While some existing neuron models reproduce experimental phenomena using such amalgamated representations of ion channels [10,12], these ion channel models are physically unrealistic because they do not contain representations of all the gating properties of the underlying isoforms. Thus, electrophysiological characteristics may be reproduced by incorrect mechanisms in these neuron models.

We created a library of 12 isoform-specific ion channel models that represent the available experimental patch clamp data. Many disease states alter the expression [37] and dynamics [38] of specific ion channel isoforms; thus, models of neurons that include ion channel models from our library can mechanistically interrogate the impact of isoform-specific alterations due to disease. Additionally, accurate ion channel modeling will improve our mechanistic understanding of the effects of different therapies that target specific ion channel isoforms.

Our library of ion channel models is not exhaustive, and we developed models only for the ion channels that are present in C-fibers [3]. However, many other ion channel isoforms exist and likely are responsible for the varied conduction responses of other neuron types [39]. Additionally, we assumed that channel isoforms retained consistent dynamics across cultured neurons, cutaneous axons, and autonomic axons; however, the dynamics of these isoforms depend on the existence and expression of particular subunits which exist only in subsets of neurons [40]. Therefore, the ion channel library and resulting C-fiber models encompass the channel dynamics required to reproduce biological responses, but do not reflect subunit-specific properties. Finally, we chose to use a standard Q10 factor to scale ion channel kinetics across different temperatures because data rigorously characterizing the Q10 factor for all of our modeled ion channel isoforms were unavailable. However, different ion channel isoforms may have different Q10 scaling factors. Nevertheless, we developed a repeatable method to fit a generic equation to patch clamp data from single ion channel characterization, which can be used to create other isoform-specific models and improve our ion channel models as additional data becomes available.

In addition to robust ion channel models, accurate neuron models require understanding of ion channel expression in different parts of the neuron. In C-fiber axons, the ion channel expression is unknown and cannot be directly measured. Therefore, we developed an automated method for tuning the expression of ion channels and pumps to reproduce accurately measurable electrophysiological responses. The PSO method was robust and generated different types of C-fiber models with very different conduction responses using the same ion channel models. Conduction responses that characterize the action potential were controlled by a small subset of ion channels, but complex conduction responses that characterize the afterpotential required many ion channels. Therefore, previous models could reproduce conduction responses that characterize the action potential fairly well (Fig 3A–3E). However, to model accurately C-fiber conduction responses during the afterpotential (Fig 3F and 3G), our PSO methodology is needed because there are complex interaction between ion channels.

Even though our PSO was robust and accurately determined two different types of C-fiber models, there were limitations with the PSO. We created a library of ion channel models, but comparable data characterizing the NaK pump and NaCa exchanger were not available, and we implemented published pump models in our C-fibers [13,14]. We also assumed a uniform distribution of each ion channel density along the length of the axon, which may not reflect differences in distributions near the axon initial segment or axon terminals. However, many of the applications of our C-fiber model interface with the axons where ion channel densities are likely more uniform [41]. Long-term conduction responses such as ADS and the recovery cycle depended heavily on the NaK pump current, and updated ion pump models may improve the accuracy of our models in reproducing these responses. Long-term conduction responses also require long simulation times: therefore, we used ADS as a validation metric following the PSO (~12 hours per ADS simulation), and we simplified the recovery cycle characterization. Specifically, during the PSO, we simulated the recovery cycle using five and seven simulations for autonomic and cutaneous C-fibers, respectively, rather than the 125 simulations used to characterize the full recovery cycle. As a result, each PSO generation completed in ~30 minutes, but when we simulated the complete recovery cycle for the final models and characterized the sensitivity of the recovery cycle to changes in each model parameter, we found that a model that does not meet all the criteria defined by our feature selection may outperform a model that does (Fig 6). Therefore, future PSO implementations that generate other models must also balance computational speed and model accuracy. Finally, the PSO-defined C-fiber models, across the range of fiber diameters, did not always account for the full range of experimentally recorded conduction responses. C-fibers with different ion channel densities likely account for the biological variability seen in experimental conduction responses. Thus, future work could use our PSO approach to define a population of autonomic and cutaneous C-fiber models that account for the full range of biological variability.

Our framework for creating neurons models with the PSO and withheld testing data is a robust approach to reverse engineer neuron models when direct measurements of ion channel and pump expression data are unavailable. While we developed two mammalian C-fiber models with our PSO, the methods can be extended to optimize objectively other types of neuron models. However, additional validation of the PSO must be conducted to ensure that the approach is sufficiently robust to model accurately other types of neurons. To develop models of other neuron types, researchers simply define which parameters to optimize in the PSO and update the model evaluation module to simulate the conduction responses of interest and compare to experimental data. The PSO can then optimize the model parameters of any neuron model to match the desired conduction responses. However, running the PSO many times will not always results in the same set of model parameters. Therefore, researchers should simulate multiple PSO runs and withhold a subset of data to test the models after optimization to

determine the most robust model. We used ADS for the post-PSO testing given that its long simulation time would make the PSO prohibitively slow, and ADS enabled evaluation of performance across many ion channels given the complexity and time course of the response. The PSO method is most limited by the amount and variety of electrophysiological data to use as performance criteria.

Accurate C-fiber models enable optimization of stimulation therapies and advancement of our mechanistic understanding of autonomic regulation, touch sensations, and pain. A broad range of devices—termed bioelectronic medicines—are envisioned for treatment of disease using electrical stimulation and block of peripheral nerve fibers [42–44]. The targeted nerves are composed primarily of C-fiber afferents, and several therapies target stimulation or block of specific C-fibers. Thus, our optimized models of autonomic and cutaneous C-fibers are important tools for the design, analysis, and optimization of electrical nerve stimulation therapies. Beyond developing new therapies, our C-fiber models can be used to understand the physiology of pain. Previous modeling work has shown that action potentials in C-fibers can fail to propagate through T-junctions [12]. The model used in this study combined many ion channel isoforms into a single ion channel and did not account for all the ion channels present in cutaneous C-fibers. Therefore, our C-fiber models, which better reproduce experimental data, can similarly be used to assess pain transmission and, by altering the expression of different ion channels in silico, mechanistically determine which ion channel isoforms are responsible for spike failure. Additionally, the isoform-specific approach will improve our understanding of pain by allowing quantification of C-fiber responses in various pathophysiological states with known isoform-specific alterations [45].

## 4 Methods

We used NEURON v7.6.2 to simulate biophysically-realistic axon models [46]. We developed the PSO framework and conducted all data analysis in MATLAB R2019a (MathWorks, Inc., Natick, MA). We ran all simulations on the Duke Compute Cluster. All simulations were conducted at 37˚C unless otherwise indicated. All code and data for the C-fiber models, for the PSO framework, and to reproduce the figures in this paper are available at https://gitlab.oit.duke.edu/bjt20/thio_cfiber.

### Ion channel modeling

We developed models of specific ion channel isoforms found in unmyelinated peripheral neurons [3] using voltage clamp data from multiple publications. The ion channel models constituted basis functions, fundamental building blocks that can combine to describe a complex system, for our C-fiber models. All equations, descriptions, references, and metadata are available in S1 Table. Using the MATLAB Curve Fitting Tool, we fit steady-state and time constant gating parameter data from patch clamp recordings of specific ion channel isoforms transfected into or native to cultured cells. For each channel, we allowed for two activation gates ("fast" and "slow") and three inactivation gates ("fast", "slow", and "ultra-slow") which do not have specific definitions, but rather refer to the relative difference between kinetics when multiple gating mechanisms were clearly indicated in literature data; most channels were well-represented by a subset of the five types of gates. We determined equations of steady-state activation (m, n) and inactivation (h, s, u) gating variables by fitting Eq 1 to experimental data:

$$m, n, h, s, u = \left[ \frac{1-c}{1 + e^{(v-a)/b}} + c \right]^k \qquad \text{Eq1}$$

where v is the transmembrane potential (mV), k is defined by the user, and the other

parameters (a, b, c) were obtained by fitting. For inactivation gates (h, s, u), k = 1 and b > 0, and for activation gates (m, n), k = 1/3 and b < 0. Activation is typically better modeled using a multi-exponential term for the activation gating variable (e.g., $m^3$) [16]. The correct exponential power k for this term is not known for every ion channel; therefore, we used k = 1/3 for all channels in anticipation of a cubic gating term for activation.

We determined equations for time constants of activation ($\tau_m$, $\tau_n$) and inactivation ($\tau_h$, $\tau_s$, $\tau_u$) by fitting Eq 2 to experimental data:

$$\tau_m, \tau_n, \tau_h, \tau_s, \tau_u = \frac{a}{e^{(v-b)/c} + e^{(v-d)/f}} + g + \left(\frac{x}{1 + e^{(v-y)/z}}\right) \qquad \text{Eq2}$$

where v is the transmembrane potential (mV) and the other parameters (a, b, c, d, f, g, x, y, z) were obtained by fitting. For all channels, a > 0, c > 0, f < 0. We only used the g and the parenthetical term when time constant data at one or both extremities of v clearly trended toward a non-zero, horizontal asymptote, corresponding to gates with slow or incomplete inactivation (e.g., $\tau_h$ in Fig 1). This fitting method consistently resulted in activation being too slow by a factor of approximately two when compared to experimental activation curves. This was likely caused by applying monoexponential fits of time constants to cubic time courses of activation; therefore, we adjusted the time constant fits for activation gates by dividing the whole equation by a factor of two, which improved responses (S1–S11 Figs). To account for the scaling of ion channel kinetics at different temperatures, we used a standard Q10 factor of 3, which is within the range (1.5 to 4) of Q10 factors used in previous C-fiber modeling studies [11–14].

## Ion Concentrations, Diffusion, and Accumulation

We simulated dynamic changes in the reversal potential of each ion calculated at each time step using the intracellular and periaxonal ion concentrations. We used the $Ca^{2+}$ accumulation, buffering, pump, and diffusion mechanisms defined in [13], and we used the $Na^+$ and $K^+$ accumulation and diffusion mechanisms defined in [11] (S1 Table). Both accumulation mechanisms allowed diffusion between the periaxonal space (30 nm thick [47]) and an extracellular bath with constant ion concentrations. To prevent model errors due to unrealistically low ion concentrations, we imposed a minimum concentration of 100 nM. Finally, we implemented a set of balancing channels and pumps that allowed us to control the transmembrane potential [11]. The balance channel and pump created either a leakage channel or pump for each ion ($Na^+$, $K^+$, and $Ca^{2+}$) such that the net flow for each ion was zero at the rest potential.

## Model geometry

We combined the ion channel models, pumps, and accumulation mechanisms into a spatially extended unmyelinated C-fiber axon model (Fig 8A). We modeled the C-fiber as a 5 mm long straight cylinder with a 30 nm periaxonal space [48]. We ran all simulations with a compartment length of 8.33 μm and time step of 5 μs [3]. In optimization, we simulated conduction responses for only 1 μm diameter C-fibers. However, when characterizing the fibers, we included simulations that span the range of C-fiber diameters (0.5–1.5 μm) [49, 50].

## Particle swarm optimization framework

We used multi-objective particle swarm optimization to determine 17 parameters of our 1 μm C-fiber models: the maximum conductances of the 12 ion channels, maximum currents of the NaK pump and NaCa exchanger, intracellular resistance, rest potential, and the total conductance and pump current (Fig 8B). We assumed the membrane capacitance was ~1.33 μF/cm$^2$

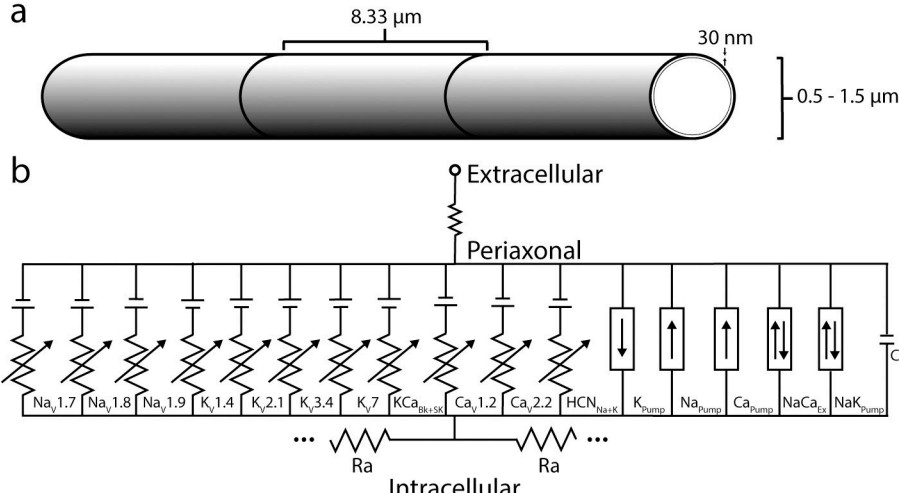

**Fig 8. Geometry and electrical properties of C-fiber models.** a) Fiber geometry with 8.33 μm-long segments and 30 nm periaxonal space. b) Cable model representation of a single segment, showing ion channels and pumps present in the C-fiber membrane.

[13] and leakage currents were controlled by the balance function. There are five steps in the PSO: initialization, evaluation, election, update, and termination (Fig 2).

In the initialization step, we defined 40 particles, where each particle was a candidate C-fiber with location defined by 17 log-uniform randomly initialized parameter values and a log-uniform randomly initialized velocity. Except for rest potential and internal resistance, particle dimension values were log-spaced, unitless values bounded between -6 and 0. We used a logarithmic representation because it allowed for differences of several orders of magnitude between parameter values. Throughout the PSO, rather than defining the maximum conductances or currents of the channels and pumps directly, we normalized the particle's ion channel and pump parameters by the sum of all ion channel and pump parameters. We controlled the sum of all currents and conductances using a single parameter bounded from 0.1 to 5 mS/cm$^2$ (or mA/cm$^2$ for maximum currents) and used this value along with the normalized channel and pump specific parameter values to calculate each channel or pump's maximum conductance (Eq 3) or current (Eq 4):

$$\bar{g}_{\text{ion channel}} = \text{total particle values} * \frac{\text{ion channel particle value}}{\sum \text{all ion channel particle values}} \qquad \text{Eq3}$$

$$I_{\text{max}} = \text{total particle values} * \frac{\text{pump particle value}}{\sum \text{all ion channel particle values}} \qquad \text{Eq4}$$

This approach allowed the PSO to change a single parameter and create proportional changes to all parameter values, analogous to adjusting the total membrane resistance. Adjusting total membrane resistance explicitly proved useful for matching threshold-related criteria. Only the intracellular resistance and rest potential did not undergo the normalization process. Instead, the intracellular resistance was bound from 10 to 100 Ω, comparable to other fiber models [10–12], and the rest potential was bound from -50 to -78 mV, consistent with experimental recordings [51,52]. Finally, we selected each particle's initial velocity from a uniform distribution from 0 to 10% of the range between search bounds for a particular dimension (see Methods section "Computing Fitness").

In the evaluation step, we determined the fitness of each particle by quantifying seven target performance criteria for the autonomic C-fiber model and six target performance criteria for the cutaneous C-fiber model: conduction velocity, chronaxie, action potential duration, existence of a short-duration supernormal period (decreased threshold following an action potential), existence of a long-duration subnormal period (increased threshold following an action potential), timing of the transition from the supernormal period to the subnormal period (autonomic fiber only), existence of an absolute refractory period (autonomic fiber only), and intracellular activation threshold (cutaneous fiber only). The simulations for each metric are detailed in the Methods section "Computing Fitness". We assigned each particle a binary and analog set of fitness scores. The binary fitness score was a set of 6 (cutaneous) or 7 (autonomic) binary numbers denoting if the particle fulfilled each of the outcomes or not, as compared to the range of experimental recordings (Table 1). The analog fitness score was a set of 6 (cutaneous) or 7 (autonomic) analog numbers denoting the distance between the mean of the target range and the computed value for each outcome. In the analog fitness score, we also added a penalization term (+9) for each component of the fitness score that was not met by the binary outcome.

In the election step, we identified 7 (cutaneous) or 8 (autonomic) best particles that had the lowest value for each analog fitness score or had the overall lowest analog fitness score; the number of "leading" particles reflects the number of target performance criteria, plus one for the best overall particle. The leaders served to influence the particle values of the next generation. Therefore, including the penalization term in the analog fitness score for particles that did not meet the binary outcome was important to ensure that particles with fitness values within our target range were chosen as leaders. We randomly grouped the remaining (non-leading) particles into 7 (cutaneous) or 8 (autonomic) groups where we used each leader to attract the other particles between generations in the update step.

In the update step, we calculated the velocity of each particle using a random exploration term, the location of the group leader, and the current location of the particle:

$$\text{Velocity} = \text{exploration} + \text{exploitation} \qquad \text{Eq5}$$

$$\text{exploration} = e^{-0.015 * \text{gen}} * (2 * \text{rand}(1) - 1)) \qquad \text{Eq6}$$

$$\text{exploitation} = 0.13 * \text{rand}(1) * (\text{leader} - \text{Current Loc}) \qquad \text{Eq7}$$

where gen is the current iteration of the PSO, rand is a uniformly distributed random number between 0 and 1, leader is the normalized parameter value of the leader, and Current Loc is the normalized parameter value of the particle of interest. The tradeoff between exploitation (movement towards the leader) and exploration (new search areas) favored exploration in early generations and exploitation in later generations; the coefficient (-0.015) in the exponent of the exploration term determines the rate at which the balance changes to favor exploitation. The coefficient of the exploitation term limited the influence of the leader particle to be at most 13% of the distance between the leader and any other particle. The resulting velocity was added to the corresponding dimension of each particle to update its value (maximum conductances, maximum currents, internal resistance, and rest potential).

After updating particle values with their new velocities, the PSO looped back to the evaluation step until a termination criterion was met: the PSO was terminated when either a particle met all binary target performance criteria or the algorithm iterated through 1600 generations.

## Computing fitness

We defined the target range for each C-fiber response (Table 1) to be within 10% of the range of the published data for the autonomic C-fiber models and within the range of the published data for cutaneous C-fiber models. For binary fitness quantification, we determined if the C-fiber conduction responses for a 1 μm diameter fiber fell within the target ranges for each conduction response. For analog quantification, we calculated the magnitude of the difference between the C-fiber conduction responses for a 1 μm diameter fiber and the mean of the target ranges for each conduction response. We simulated the fiber at the temperature matching the experimental measurement (Table 1), and we defined an action potential as a rising edge past 0 mV.

Conduction velocities for autonomic fibers were targeted from 0.45 to 1.76 m/s [23, 24] while cutaneous C-fibers were targeted from 0.5 to 1.5 m/s [25]. We determined the conduction velocity by activating the axon model with a current injection at one end and recording the time required for the action potential to propagate from 1.25 to 3.75 mm along the axon.

We defined the target range for chronaxie from 750 to 1500 μs for the autonomic fiber [26, 27] and from 500 to 1500 μs for the cutaneous fiber [28, 29]. The chronaxie is the shortest pulse width that can evoke an action potential with twice the rheobase current, where the rheobase is the minimum current needed to evoke an action potential with a pulse of infinite duration. We simulated activation thresholds using a monopolar point source placed 0.15 mm [53, 54] from the center of the axon and multiple monophasic cathodic pulses (pulse width = PW = 0.02, 0.035, 0.05, 0.075, 0.1, 0.2, 0.5, 1, 2, 10 ms). We determined the chronaxie ($T_{ch}$) by fitting the activation thresholds ($I_{th}$) to Eq 8:

$$\log_{10}(I_{th}) = \log_{10}\left[I_{rh} * \left(1 + \frac{T_{ch}}{PW}\right)\right] \qquad \text{Eq 8}$$

where $I_{rh}$ is the rheobase current.

For action potential duration, we targeted a range of 1.35 to 3 ms for the autonomic C-fiber [30, 31, 33] and 1.5 to 2.6 ms at 24°C for the cutaneous C-fiber [32]. During the autonomic conduction velocity simulation, we recorded the transmembrane potential of the middle compartment to obtain the voltage trace used in determining the action potential duration. We also ran a second conduction velocity simulation at 24°C for cutaneous C-fibers to record the transmembrane potential.

For recovery cycle quantification, we used intracellular current injection at the center of the fiber and quantified the baseline threshold for a single 100 μs anodic pulse to evoke an action potential recorded at the end of the fiber. We then delivered a pair of 100 μs pulses and examined the response to the second pulse at different ISIs, including its change in threshold relative to the baseline threshold. The first pulse was delivered at 1.5x threshold. We defined four separate objectives to characterize the autonomic C-fiber recovery cycle. First, target refractory period was fulfilled if the threshold for the second pulse was >100% baseline threshold for ISI = 13 ms. Second, the target supernormal period was fulfilled if the second pulse evoked an action potential at 90% baseline threshold but not at 70% baseline threshold for ISI = 23 ms. Third, the target subnormal period was fulfilled if the threshold for the second pulse was increased 2 to 15% above baseline for ISI = 250 ms. Finally, the target timing of the transition from supernormality to subnormality was fulfilled if it occurred between ISIs of 50 and 100 ms, i.e., if the second pulse at baseline threshold amplitude elicited an action potential at ISI = 50 ms but not at ISI = 100 ms [33]. We quantified an analog transition as the mean percent difference between baseline threshold and paired pulse threshold at ISI = 50 and 100 ms.

For the recovery cycle of the cutaneous C-fiber, we implemented intracellular stimulation at the center of the fiber and evaluated the change in threshold for the second pulse, relative to baseline, for ISI = [5, 10, 15, 20, 25, 30, 150 ms] at 33˚C. The target "supernormal" period was fulfilled if the change in threshold was <10% for at least one short ISI (5 to 30 ms) and the threshold at all time points exceeded 90% of the baseline threshold. The target subnormal period was achieved if the threshold for the second pulse at ISI = 150 ms was increased by 10–30% relative to baseline.

Finally, we determined the threshold of the cutaneous fiber by using an intracellular current injection at one end of the fiber with a 10 ms anodic pulse at 24˚C. We detected action potential propagation at the center of the C-fiber model. The current was injected into the end of the fiber to mimic the bleb injection performed experimentally [52]. The target was met if the baseline current threshold was between 15 and 150 pA.

## Activity-dependent slowing

To quantify activity-dependent slowing (ADS), we increased the length of the models to 50 mm and the compartment length to 100 μm to improve the resolution of the changes in conduction velocity through increased conduction distance without increasing the simulation time (12 h for a single ADS simulation). These modifications caused a change of ~1% in the conduction velocity of a single spike. We simulated ADS by stimulating the model for 180 s with pulses at 0.5, 1, 2, and 4 Hz [4, 55, 56]. Each pulse was a 0.5 ms voltage clamp, which set the voltage of the second segment in the model to 0 mV. We used a voltage clamp instead of a current injection because the current threshold is also activity-dependent, and if the magnitude of the delivered current was too large, the solver returned not-a-number in cases where an ion concentration became negative. We recorded the latencies at which the action potential propagated through the segments located 1.25 and 3.75 cm distal to stimulation and then calculated the corresponding conduction velocity.

## Sensitivity analysis

To quantify the impact of each model parameter on the conduction responses, we conducted a sensitivity analysis where we simulated each conduction response when each model parameter value was increased or decreased by 50%. We quantified the resulting percent change in each conduction response except for recovery cycle. For recovery cycle, we quantified the percent change from the optimized model as the error between the experimental data from literature [32,33] and the model. Specifically, we quantified the error between the experimental data and model using the root mean square (RMS) deviation between the percent change in threshold determined by the experimental data and a linear interpolation of our model for corresponding ISI values.

## Model reduction

To determine the model parameter needed to produce an accurate model, we sequentially set model parameters to zero and computed each conduction response. The order of removing model parameters is important for determining which model parameters are necessary to reproduce accurately the experimental data: we removed model parameters based on the mean magnitude of the conduction responses' sensitivities to changes in each model parameter (±50%) from lowest to highest. We then quantified if a model parameter was necessary or not by comparing the conduction responses from the resulting models to the experimental ranges that we defined to compute the fitness of a C-fiber models. If a simulated conduction response remained in the experimental range after removing a model parameter, we determined that

the model parameter was unnecessary. However, if the conduction response no longer remained in the experimental range after removing a model parameter, we determined that the model parameter was necessary.

## Supporting information

**S1 Table. Ion Channel Metadata.**
(XLSX)

**S2 Table. Sensitivity of each autonomic C-fiber conduction response (%) to changes in membrane parameter values.** Sensitivity values with magnitude <0.1 are marked as 0.
(XLSX)

**S3 Table. Sensitivity of each cutaneous C-fiber conduction response (%) to changes in membrane parameter values.** Sensitivity values with magnitude <0.1 are marked as 0.'
(XLSX)

**S1 Fig. Comparison of our isoform-specific model of $Na_V 1.7$ (solid line; black except in panel A) to the Schild et al. 1997 model (dash-dotted line; red except in panel A)** [14]**, Tigerholm et al. 2014 model (dashed line; blue except in panel A)** [11]**, Sundt et al. 2015 model (dotted line; orange except in panel A)** [12]**, and experimental data (markers)** [57–61]. A) steady-state gating parameter values, B) fast-activation time constant, C) fast-inactivation time constant, D) slow-inactivation time constant. Panels without a solid line indicate that gating parameter was not included in our isoform-specific ion channel model. E) Fast-activation time constant and F) fast-inactivation time constant at different temperatures.
(TIF)

**S2 Fig. Comparison of our isoform-specific model of $Na_V 1.8$ (solid line; black except in panel A) to the Schild et al. 1997 model (dash-dotted line; red except in panel A)** [14]**, Tigerholm et al. 2014 model (dashed line; blue except in panel A)** [11]**, and experimental data** [17, 18, 20, 21, 62, 63]. A) steady-state gating parameter values, B) fast-activation time constant, C) fast-inactivation time constant, D) slow-inactivation time constant, E) Ultra-slow-inactivation time constant. Panels without a solid line indicate that gating parameter was not included in our isoform-specific ion channel model. F) Fast-activation time constant, G) fast-inactivation time constant, and H) slow-inactivation time constant at different temperatures.
(TIF)

**S3 Fig. Comparison of our isoform-specific model of $Na_V 1.9$ (solid line; black except in panel A) to the Tigerholm et al. 2014 model (dashed line; blue except in panel A)** [11] **and experimental data** [64–68]. A) steady-state gating parameter values, B) fast-activation time constant, C) fast-inactivation time constant, D) slow-inactivation time constant. Panels without a solid line indicate that gating parameter was not included in our isoform-specific ion channel model. E) Fast-activation time constant, F) fast-inactivation time constant, and G) slow-inactivation time constant at different temperatures.
(TIF)

**S4 Fig. Comparison of our isoform-specific model of $K_V 1.4$ (solid line; black except in panel A) to the Schild et al. 1997 model (dash-dotted line; red except in panel A)** [14] **and experimental data** [69–73]. A) steady-state gating parameter values, B) fast-activation time constant, C) fast-inactivation time constant, D) slow-inactivation time constant. Panels without a solid line indicate that gating parameter was not included in our isoform-specific ion

channel model. E) Fast-activation time constant, F) fast-inactivation time constant, and G) slow-inactivation time constant at different temperatures.
(TIF)

**S5 Fig.** Comparison of our isoform-specific model of $K_V2.1$ (solid line; black except in panel A) to the Schild et al. 1997 model (dash-dotted line; red except in panel A) [14], Tigerholm et al. 2014 model (dashed line; blue except in panel A) [11], Sundt et al. 2015 model (dotted line; orange except in panel A) [12], and experimental data [74–78]. A) steady-state gating parameter values, B) fast-activation time constant, C) fast-inactivation time constant. Panels without a solid line indicate that gating parameter was not included in our isoform-specific ion channel model. D) Fast-activation time constant and E) fast-inactivation time constant at different temperatures.
(TIF)

**S6 Fig. Comparison of our isoform-specific model of $K_V3.4$ (solid line; black except in panel A) to the Tigerholm et al. 2014 model (dashed line; blue except in panel A) [11] and experimental data** [79–81]. A) steady-state gating parameter values, B) fast-activation time constant, C) fast-inactivation time constant. Panels without a solid line indicate that gating parameter was not included in our isoform-specific ion channel model. D) Fast-activation time constant and E) fast-inactivation time constant at different temperatures.
(TIF)

**S7 Fig. Comparison of our isoform-specific model of $K_V7$ (solid line; black except in panel A) to the Schild et al. 1997 model (dash-dotted line; red except in panel A) [14], Tigerholm et al. 2014 model (dashed line; blue except in panel A) [11], Sundt et al. 2015 model (dotted line; orange except in panel A) [12], and experimental data** [78, 82–85]. A) steady-state gating parameter values, B) fast-activation time constant, C) slow-activation time constant. Panels without a solid line indicate that gating parameter was not included in our isoform-specific ion channel model. E) Fast-activation time constant and H) slow-activation time constant at different temperatures.
(TIF)

**S8 Fig. Responses of KCa ion channel.** ABDE) Responses of Bk subunit of the KCa channel [86–89]. A) Steady-state activation gating properties as a function of patch clamp voltage and intracellular calcium concentration. B) Steady-state interaction between activating and inhibiting gating properties as a function of patch clamp voltage and intracellular calcium concentration. D) Time constant for activation as a function of patch clamp voltage and intracellular calcium concentration. E) Time constant for inhibition as a function of patch clamp voltage and intracellular calcium concentration. CF) Responses of Sk subunit of the KCa channel [90–92]. C) Steady-state gating properties as a function of intracellular calcium concentration. F) No literature data are available for the time constant of activation as a function of intracellular calcium concentration. G) Fast-activation time constant of the Bk subunit, H) fast-inactivation time constant of the Bk subunit, and I) fast-activation time constant of the Sk subunit at different temperatures.
(TIF)

**S9 Fig. Comparison of our isoform-specific model of $Ca_V1.2$ (solid line; black except in panel A) to the Schild et al. 1997 model (dash-dotted line; red except in panel A) [14] and experimental data** [93–97]. A) steady-state gating parameter values, B) fast-activation time constant, C) fast-inactivation time constant, D) slow-inactivation time constant. Panels without a solid line indicate that gating parameter was not included in our isoform-specific ion

channel model. E) Fast-activation time constant and F) fast-inactivation time constant at different temperatures.
(TIF)

**S10 Fig. Comparison of our isoform-specific model of Ca$_V$2.2 (solid line; black except in panel A) to the Schild et al. 1997 model (dash-dotted line; red except in panel A) [14] and experimental data [98–107].** A) steady-state gating parameter values, B) fast-activation time constant, C) fast-inactivation time constant, D) slow-inactivation time constant. Panels without a solid line indicate that gating parameter was not included in our isoform-specific ion channel model. E) Fast-activation time constant, F) fast-inactivation time constant, and G) slow-inactivation time constant at different temperatures.
(TIF)

**S11 Fig. Comparison of our isoform-specific model of HCN (solid line; black except in panel A) to the Tigerholm et al. 2014 model (dashed line; blue except in panel A) [14] and experimental data [108–112].** A) steady-state gating parameter values, B) fast-activation time constant, C) slow-activation time constant. Panels without a solid line indicate that gating parameter was not included in our isoform-specific ion channel model. D) Fast-activation time constant and E) fast-activation time constant at different temperatures.
(TIF)

**S12 Fig. Convergence plots for PSO runs that matched experimental ADS.** A) Convergence of autonomic C-fiber PSO. B) Convergence of cutaneous C-fiber PSO. The red numbers indicate the number of performance criteria met by the best particle in the swarm.
(TIF)

**S13 Fig. Activity dependent slowing for all PSO generated C-fiber models.** A) Autonomic and B) Cutaneous C-fiber models are shown compared to experimental data for all PSO runs. The red lines indicate the 2 C-fiber models used in all other analyses.
(TIF)

**S14 Fig. Intracellular Na$^+$ and periaxonal K$^+$ concentrations of the model C-fibers during activity-dependent slowing.** A) Autonomic and B) cutaneous C-fiber ion concentrations are shown with % changes in conduction velocity for 1 μm C-fibers during 4 Hz stimulation.
(TIF)

## Acknowledgments

We would like to thank Eric Musselman for helping implement the other C-fiber models from literature. We would like to thank Gene Yu and Bradley Barth for help with figure design. We would like to thank the Duke Research Computing for housing and maintaining the computational resources used to run the simulations in this study.

## Author Contributions

**Conceptualization:** Brandon J. Thio, Nathan D. Titus, Nicole A. Pelot, Warren M. Grill.

**Data curation:** Brandon J. Thio, Nathan D. Titus, Nicole A. Pelot.

**Formal analysis:** Brandon J. Thio, Nathan D. Titus.

**Funding acquisition:** Nicole A. Pelot, Warren M. Grill.

**Investigation:** Brandon J. Thio.

**Methodology:** Brandon J. Thio, Nathan D. Titus, Nicole A. Pelot.

**Project administration:** Nicole A. Pelot.

**Software:** Brandon J. Thio, Nathan D. Titus.

**Supervision:** Nicole A. Pelot, Warren M. Grill.

**Validation:** Brandon J. Thio, Nathan D. Titus.

**Visualization:** Brandon J. Thio, Nathan D. Titus.

**Writing – original draft:** Brandon J. Thio.

**Writing – review & editing:** Brandon J. Thio, Nathan D. Titus, Nicole A. Pelot, Warren M. Grill.

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
