## [Decision Letter · Decision Letter 0]

10 Jul 2024

Dear Dr Grill,

Thank you very much for submitting your manuscript "Reverse-engineered models reveal differential membrane properties of autonomic and cutaneous unmyelinated fibers" for consideration at PLOS Computational Biology.

As with all papers reviewed by the journal, your manuscript was reviewed by members of the editorial board and by several independent reviewers. In light of the reviews (below this email), we would like to invite the resubmission of a significantly-revised version that takes into account the reviewers' comments.

We cannot make any decision about publication until we have seen the revised manuscript and your response to the reviewers' comments. Your revised manuscript is also likely to be sent to reviewers for further evaluation.

Sincerely,

Bruce Graham

Guest Editor

PLOS Computational Biology

Lyle Graham

Section Editor

PLOS Computational Biology

Reviewer's Responses to Questions

**Comments to the Authors:**

Reviewer #1: This is a very interesting paper aimed at studying the electric properties of unmyelinated C-fiber axons. Using a library of isoform-specific ion channel models and a particle swarm optimization framework, the authors were able to reproduce i) the experimentally-observed conduction velocity, ii) the time necessary to excite the fiber, iii) the duration of action potential, iv) the spike threshold and v) the paired-pulse recovery duration. This method could be used in other thin unmyelinated axons.

The paper is very well conducted and written. I have only relatively minor remarks.

1) Please, briefly define chronaxie when it first appears.

2) A few panels are empty in most of Supplementary Figures. Please correct or delete them.

Reviewer #2: This article reports the development of a novel method called particle swarm optimization (PSO) framework to optimize the computer models of experimentally inaccessible membrane properties and investigates the applicability of this novel method to unmyelinated C-fibers of autonomic and cutaneous pain-related nerves where direct patch-clamp recording from thin axons are practically impossible. Since the authors found that the unmyelinated C-fiber models reproduced not only the conduction responses of those observed experimentally but also phenomenon not used for optimization, the authors conclude this reverse engineering approach with PSO algorithm would have the potential to generate other neurons where voltage clamp data are not available. Strengths include systematic exploration with varying a set of voltage-sensitive channel isoforms and maximum conductance/pump-current levels by repeated calculation until the criteria are met.

Major concerns:

It is not clear how the authors chose a set of voltage-gated channels on the C-fibers. Do all channels have a rationale with immunohistochemistry or related morphological studies? Is it rational to include the channels with no evidence for their existence? What is an assumption on the passive membrane properties and intracellular environment? How they defined periaxonal space? At least, the authors need to thorough description of these points.

It is also not clear whether they assume a uniform distribution of the set of channels for the whole courses of the axons. It is well known that distribution is not uniform, i.e. highest Na conductances at the proximal axons near the physiological spike initiation sites.

In my opinion, testing in other models of the axon with experimental voltage-clamped data, such as the squid giant axon or hippocampal mossy fiber axon, is obligately to show the rationality of this optimization approach.

Reviewer #3: Major comments

1. All but two of the sources of experimental data for the ion channel properties were performed at or near room temperature (Table S1). It would be helpful in the Methods section to comment on how the temperature-dependence (Q10) of each of the channels was determined/arrived at and their validity used in the mod files. A supplemental figure illustrating the temperature dependence of the conductances/pumps on the voltage-gated currents’ kinetics would be instructive. Also, it is not clear how the PSO was applied between two different temperatures when optimizing action potential duration and intracellular current threshold whose target temperatures were near room temperature (Table 1).

2. Supplemental Figures 1-11 are a bit difficult, especially in the context of demonstrating that they “accurately represent the available patch clamp data”. The work presented here might be better served by simply comparing their new channel models to the experimental data, rather than the channel models used in previous in silico studies. It also might be here that the temperature-dependence is illustrated.

Minor comments

1. Table 1 the range of values determined in this study might be added to this table for comparison to the “Target” / experimentally measured ranges. And so might want to move this table to the results section.

2. Page 5, line 83: Please clarify the sentence “We selected isoforms such that each class of identifiable voltage-sensitive current in peripheral C-fibers was represented by at least one model ion channel.”

3. Page 5, line 85 : How were the channel models validated? This goes to major comment #1 above. What is meant by “basis functions”?

4.Page 8, line 108: What is meant by simulating 50 and 20 PSOs? Does this refer to different sets of initial conditions? If so, why different values for the two types of axons?

5. Page 10, Figure 3, Panels A & B: The authors should comment on why values for conduction velocity and action potential duration determined by SPO did not account for the full range of experimentally-recorded values.

6. Page 14, line 193: Suggest renaming the heading to something like “Specific Ion channels necessary to model C-fiber electrogenesis”

7. Page 18, line 253: The authors state that the 12 channel models “accurately represent the experimental patch clamp data”. It appears that in Figures S1-11 the models deviate from the experimentally-recorded values in a number of cases.

8. Page 22, line 360: Might be useful to report the conductances/current density for the balance channel/pump as it reports membrane resistivity.

**Have the authors made all data and (if applicable) computational code underlying the findings in their manuscript fully available?**

Reviewer #1: Yes

Reviewer #2: Yes

Reviewer #3: Yes

PLOS authors have the option to publish the peer review history of their article (what does this mean?). If published, this will include your full peer review and any attached files.

Reviewer #1: No

Reviewer #2: No

Reviewer #3: No
---

## [Editor Report · Decision Letter 1]

11 Sep 2024

Dear Dr Grill,

We are pleased to inform you that your manuscript 'Reverse-engineered models reveal differential membrane properties of autonomic and cutaneous unmyelinated fibers' has been provisionally accepted for publication in PLOS Computational Biology.

Best regards,

Bruce Graham

Guest Editor

PLOS Computational Biology

Lyle Graham

Section Editor

PLOS Computational Biology

Please ensure that all citations of papers and links to tables / figures are correct as a number are missing in the text of the revised manuscript.

---

## [Editor Report · Acceptance letter]

2 Oct 2024

PCOMPBIOL-D-24-00176R1 

Reverse-engineered models reveal differential membrane properties of autonomic and cutaneous unmyelinated fibers

Dear Dr Grill,

I am pleased to inform you that your manuscript has been formally accepted for publication in PLOS Computational Biology. Your manuscript is now with our production department and you will be notified of the publication date in due course.

With kind regards,

Anita Estes
